# COVID-19 vaccination and leprosy–A UK hospital-based retrospective cohort study

**Barbara de Barros**[1,2]*, **Rachel Pierce**[3], **Cathryn Sprenger**[4], **Eugene Liat Hui Ong**[4], **Stephen L. Walker**[1,2,3]

**1** Hospital for Tropical Diseases, University College London Hospitals NHS Foundation Trust, London, United Kingdom, **2** Faculty of Infectious and Tropical Diseases, London School of Hygiene & Tropical Medicine, London, United Kingdom, **3** Department of Dermatology, University College London Hospitals NHS Foundation Trust, London, United Kingdom, **4** Dermatology Department, St Georges University Hospitals NHS Foundation Trust, London, United Kingdom

* Barbara.de-barros@lshtm.ac.uk

**Data Availability Statement:** The authors confirm that all data underlying the findings are fully available without restriction. All relevant data are within the paper and its Supporting Information file.

## Abstract

### Background

Individuals with leprosy are at risk of leprosy reactions, T-cell mediated immunological complications, which lead to nerve function impairment. Leprosy reactions require systemic immunosuppression which is a risk factor for severe COVID-19. Vaccination for SARS-CoV-2 infection is recommended in the UK and became widely available in 2021 with individuals at increased risk of severe disease, including the immunosuppressed, prioritised.

Vaccines for SARS-CoV-2 may provoke a T cell response. The latter poses a theoretical risk of provoking an immunological response to latent *Mycobacterium leprae* infection leading to clinical disease or in those with clinical disease triggering a leprosy reaction. BCG vaccination is associated with the development of leprosy in a small proportion of healthy contacts of people with leprosy within twelve weeks of administration. BCG causes a Th1 immune response.

### Methodology/Principal findings

We performed a retrospective cohort study to determine the SARS-CoV-2 vaccination status of individuals diagnosed with leprosy attending the Leprosy Clinic in 2021 and whether any had developed leprosy or experienced a new leprosy reaction within twelve weeks of receiving a dose of a SARS-CoV-2 vaccine. The electronic patient records were used to retrieve data.

Fifty-two individuals with leprosy attended the clinic in 2021 of which five people were newly diagnosed with leprosy. Thirty-seven (71%) were male and the median age was 48.5 years old (Range 27–85 years). Eight (15.4%) individuals were taking multi-drug therapy (MDT) and eight (15.4%) had completed MDT within three years of the study. Twenty-two (41.5%) individuals were prescribed a systemic immunosuppressant drug during 2021. Ten (18.9%) individuals have one or more risk factors for severe COVID-19. The SARS-CoV-2 vaccination status of fifty (96%) were recorded of which forty-nine were vaccinated (98%). One individual had declined vaccination.

**Funding:** The author(s) received no specific funding for this work.

**Competing interests:** The authors have declared that no competing interests exist.

One individual was diagnosed with borderline tuberculoid (BT) leprosy having developed red skin lesions with reduced sensation (which increased in size and number) and thickened peripheral nerves one week after a second dose of BNT162b2 vaccine. Another individual who had completed MDT more than three years earlier developed red plaques and tender thickened nerves consistent with a leprosy Type 1 reaction eight weeks after a single dose of BNT162b2 vaccine (having received two doses of CoronaVac vaccine three months earlier).

## Conclusions/Significance

The development of BT leprosy and a Type 1 reaction in another individual shortly after a dose of BNT162b2 vaccine may be associated with vaccine mediated T cell responses. The benefits of vaccination to reduce the risk of severe COVID-19 outweigh these unwanted events but data from leprosy endemic countries may provide further information about potential adverse effects of augmented T cell responses in individuals with leprosy or latent *M. leprae* infection.

### Author summary

Leprosy is a neglected tropical disease caused by *Mycobacterium leprae*. The clinical presentation varies depending on the immune response of the affected person. The infection is curable with antimicrobial therapy but individuals with leprosy can experience immune-mediated complications known as leprosy reactions. There are reports of vaccines, including those for SARS-CoV-2 being associated with the development of leprosy or leprosy reactions. We performed a retrospective study to determine the vaccination status of all individuals diagnosed with leprosy attending the Leprosy Clinic in 2021 and whether any individual had developed leprosy or a new leprosy reaction within 12 weeks of receiving a dose of a SARS-CoV-2 vaccine. Fifty-two individuals with leprosy attended the clinic, five newly diagnosed and eight individuals were on antimicrobial treatment. Twenty-two individuals were prescribed an immunosuppressant drug during 2021. Forty-nine individuals had at least one vaccine dose, one was unvaccinated There was no vaccination record for two individuals. Two individuals met our criteria for a SARS-CoV-2 vaccine associated new leprosy adverse event. We have reviewed other cases of vaccine associated leprosy adverse events and conclude that the benefits of vaccination to reduce the risk of severe COVID-19 outweigh these unwanted events. It is important for clinicians to be aware of the potential leprosy adverse events associated with SARS-CoV-2 vaccination and to advise leprosy affected individuals to report any new symptoms immediately.

## Introduction

Leprosy is caused by *Mycobacterium leprae and Mycobacterium lepromatosis* [1]. The clinical phenotype of leprosy exhibited by an affected individual is determined by the host immune response. The Ridley-Jopling (RJ) classification uses clinical, histopathological and bacteriological characteristics to classify the disease into: turberculoid (TT) leprosy, borderline turberculoid (BT) leprosy, borderline borderline (BB) leprosy, borderline lepromatous (BL) leprosy

and lepromatous leprosy (LL). Individuals with high cell mediated immunity to *M. leprae* produce a Th1 type immune response with granuloma formation in the presence of CD4+ cells and few or no bacteria identifiable in tissues which is characteristic of tuberculoid and borderline tuberculoid leprosy. LL is characterised by a high bacterial load, with poor granuloma formation and a predominance of CD8+ lymphocytes [2].

*M. leprae* infection is curable and since 1982 WHO have recommended the use of highly effective combinations of rifampicin, dapsone and clofazimine [3]. However, individuals with leprosy may experience immune-mediated inflammatory complications known as leprosy reactions even after successful completion of anti-microbial therapy. Leprosy reactions are associated with nerve damage and are a risk factor for leprosy associated disability [4,5]. There are two types of leprosy reactions. Type 1 reactions (T1Rs) may complicate all forms of leprosy and are characterised by the development of oedema, inflammation in pre-existing leprosy skin lesions and nerves, with pain and loss of function. T1Rs are a delayed hypersensitivity reaction to *M. leprae* antigens, which appear to be mediated via Th1 type cells expressing pro-inflammatory IFN-γ, IL-12 and oxygen free radical producer inducible nitric oxide synthase [6]. Erythema Nodosum Leprosum (ENL) or Type 2 Reaction is a multisystem complication characterised by painful cutaneous nodules, fever, arthralgia, arthritis, and neuritis. ENL is often severe and chronic [7]. The understanding of the immunology of ENL is limited [8]. There is evidence of T lymphocyte and macrophage activation and expression of mRNA for TNFα and interleukin (IL)-12 in the skin [7,8].

T1Rs require systemic immunosuppression with high dose oral corticosteroids [9]. ENL is treated with oral corticosteroids or thalidomide (if available) [10]. In individuals with severe adverse effects to corticosteroids or not responding to treatment, other immunomodulatory agents may be required.

Severe acute respiratory syndrome coronavirus 2 (SARS-CoV-2) was reported to have caused more than 401,726,441 cases causing more than 5,767,000 deaths by February 2022 [11]. Risk factors for severe COVID-19 and hospitalisation include diabetes mellitus, respiratory chronic disease and immunosuppression [12]. Measures recommended to reduce SARS-CoV2 transmission and/or severity have included social distancing, isolation, travel restrictions and vaccination. WHO issued guidelines suspending programs, surveys, active surveillance, and community activities in neglected tropical diseases (NTDs) programs, including leprosy [13]. In 2019 the number of new cases reported to WHO was 202,185 [14]. whereas 127,396 new cases were reported in 2020 [15]. This illustrates the impact of the COVID-19 pandemic on the access and provision of leprosy services [1].

Vaccination for SARS-CoV-2 infection is recommended in the UK and became widely available in 2021 with individuals at increased risk of severe disease (including the immunosuppressed) prioritised [16,17]. The three most used vaccines in the UK were the BNT162b2, mRNA-1273 and the ChAdOx1 nCoV-19. The BNT162b2 is RNA-based vaccine developed by Pfizer BioNTech, and it elicits neutralising antibodies and robust antigen-specific CD8[+] and Th1-type CD4[+] T-cell responses [18]. mRNA-1273 is an mRNA vaccine by Moderna, Inc., and has similar efficacy to BNT162b2 [19]. The ChAdOx1 nCoV-19 vaccine by Oxford University consists of replication-deficient chimpanzee adenoviral vector containing the SARS-CoV-19 spike protein gene. All three vaccines have been shown to induce T cell mediated immunity [20], and it has been shown to increase serum antibody levels to spike protein, with high levels of TNF-α and IFN-γ [21].

Leprosy or leprosy reactions can be precipitated by infections and vaccines [22–24]. Bacillus Calmette-Guérin (BCG) is a live attenuated vaccine developed to prevent tuberculosis (TB). The immune response elicited after BCG vaccination is associated with a Th1 immune response [25]. BCG may induce non-specific cross-protection against other pathogens and

protects against leprosy [25,26]. However, BCG is associated with the development of leprosy in 0.33% of healthy contacts of people with leprosy within 12 weeks of administration [24]. Smallpox vaccination was associated with the onset of leprosy reactions and the development of leprosy [27].

Vaccines for SARS-CoV-2 provoke a T cell response [20]. There is therefore a theoretical risk of SARS-CoV-2 vaccines provoking an immunological response to *M. leprae* bacilli or antigen leading to leprosy or leprosy reactions. There are reports of leprosy reactions following SARS-CoV-2 vaccines.

Leprosy is rare in the United Kingdom (UK) but individuals who have migrated from or lived for prolonged periods in leprosy endemic countries continue to be diagnosed [28]. There have been no reports of an individual acquiring leprosy in the UK since 1954 [29].

We wished to assess the uptake of SARS-CoV-2 vaccine amongst patients attending the Leprosy Clinic at the Hospital for Tropical Diseases (HTD), London, UK and to determine if there were any leprosy associated adverse effect.

## Methodology

### Ethics statement

Data collection was approved as part of a quality improvement project registered with the Infection Division of University College London Hospitals NHS Foundation Trust in accordance with the institutional policy. Consent was not required for anonymised data.

### Study design

We performed a retrospective cohort study to determine the SARS-CoV-2 vaccination status of all individuals diagnosed with leprosy attending the Leprosy Clinic in 2021 using the electronic patient records and whether any individuals had developed leprosy or a new leprosy reaction within 12 weeks of receiving a dose of a SARS-CoV-2 vaccine.

### Case definition

A leprosy associated adverse event was defined as the development of leprosy or a leprosy reaction and/or neuritis within twelve weeks of receiving a dose of a SARS-CoV-2 vaccine in an individual with no previous history of leprosy or a leprosy reaction and/or neuritis and who had not received treatment for a leprosy reaction and/or neuritis in the previous 12 weeks.

### Study setting

The HTD is a national referral centre in the UK for tropical and infectious diseases and has a dedicated Leprosy Clinic.

### Data collection and study population

All patients with a confirmed diagnosis of leprosy seen in the Leprosy Clinic in 2021 were included. Information on sex; age; date of leprosy diagnosis; MDT status in 2021; leprosy reactions in 2021; use of prednisolone, thalidomide or other immunosuppressants in 2021; SARS-CoV-2 vaccination status and comorbidities was recorded.

### Data management and analysis

All data were anonymised and entered into Excel. Descriptive statistics were performed in Excel.

## Results

Fifty-two individuals with a diagnosis of leprosy attended the clinic in 2021. Five had been diagnosed in 2021. Thirty-seven (71%) were male and the median age was 48.5 years old (Range 27–85). Eight (15.4%) individuals were taking multi-drug therapy (MDT) and eight (15.4%) had completed MDT within three years of the study. Twenty-three (44.2%) individuals were diagnosed with LL, nine (17.3%) with BL leprosy, one (1.9%) BB leprosy, 14 (26.9%) BT and one (1.9%) TT. Two individuals (3.8%) were diagnosed with pure neural leprosy. The RJ classification was not available for two individuals.

Twenty-one of 52 (40.3%) individuals were prescribed a systemic immunosuppressant drug during 2021 for leprosy reactions. Sixteen received oral prednisolone, nine thalidomide and three received other agents: benepali, apremilast and high dose clofazimine (some individuals received more than one agent during the period of study). One individual was prescribed methotrexate for an unrelated condition. Ten (19.2%) individuals had another risk factor for severe COVID-19, including one patient with HIV co-infection. The SARS-CoV-2 vaccination status of 50 patients (96%) was recorded. Forty-nine of 50 (98%) had received at least one dose of a SARS-CoV-2 vaccine in 2021. One individual had declined vaccination. Table 1 summarises the results.

**Table 1. Demographic, clinical and vaccination status of 52 patients attending the HTD Leprosy Clinic.**

| Characteristic | | Vaccinated (n = 49) | Not vaccinated (n = 1) | Sars-cov2 vaccination status unknown (n = 2) | Total (n = 52) |
|---|---|---|---|---|---|
| **Sex** | Male | 35 | 0 | 2 | 37 |
| | Female | 14 | 1 | 0 | 15 |
| | Age (median) | 48.5 years-old (range 27–85) | | | |
| **Ridley-Jopling classification** | TT | 0 | 1 | 0 | 1 |
| | BT | 13 | 0 | 1 | 14 |
| | BB | 1 | 0 | 0 | 1 |
| | BL | 9 | 0 | 0 | 9 |
| | LL | 22 | 0 | 1 | 23 |
| | PNL | 2 | 0 | 0 | 2 |
| | Unknown* | 2 | 0 | 0 | 2 |
| **MDT status** | On MDT | 8 | 0 | 0 | 8 |
| | Completed MDT ≤ 3 years | 7 | 1 | 0 | 8 |
| | Completed MDT > 3 years | 31 | 0 | 5 | 36 |
| **Reaction treatment status at any time during 2021 (n = 21)** | Treatment for T1R | 5 | 0 | 0 | 5 |
| | Treatment for T2R | 12 | 0 | 0 | 12** |
| | Treatment for neuritis | 4 | 0 | 1 | 5 |
| **Reaction medications prescribed at any time during 2021 (n = 21)** | Oral prednisolone | 16 | 0 | 1 | 17 |
| | Thalidomide | 9 | | 0 | 9 |
| | Other immunosuppressant | 3 | 0 | 0 | 3 |
| **Risk factors for severe COVID-19*** | | 10 | 0 | 0 | 10 |

*One individual received WHO multibacillary MDT at another centre

** One individual had an episode of T1R followed by T2R

*** excluding immunosuppression

**Table 2. The SARS-CoV-2 vaccination schemes of individuals seen at the Leprosy Clinic in 2021.**

| Number of SARS-CoV-2 vaccine | Vaccination schemes | Number who were offered vaccination, n = 50 (%) |
|:---:|:---|:---:|
| 0 | Unvaccinated | 1 (2%) |
| 1 | ChAdOx1 nCoV-19 single dose | 1 (2%) |
| 2 | BNT162b2 double dose | 4 (8%) |
| 2 | ChAdOx1 nCoV-19 double dose | 5 (10%) |
| 2 | mRNA-1273 double dose | 2 (4%) |
| 3 | BNT162b2 triple dose | 14 (28%) |
| 3 | ChAdOx1 nCoV-19 double dose plus BNT162b2 booster | 11 (22%) |
| 3 | ChAdOx1 nCoV-19 double dose plus mRNA-1273 booster | 3 (6%) |
| 3 | ChAdOx1 nCoV-19 double dose plus unknown booster | 1 (2%) |
| 3 | CoronaVac double dose plus BNT162b2 booster | 1 (2%) |
| Unknown | Vaccinated but unknown vaccine scheme | 7 (14%) |

Two males had a new leprosy event within 12 weeks of a dose of SARS-CoV-2 vaccine.

Table 2 details the types of vaccines administered. One of 50 (2%) received only one dose of ChAdOx1 nCoV-19, 11 (22%) had two doses of a SARS-CoV-2 vaccine and 30 (60%) had received three doses. Seven (14%) were vaccinated but the type and number were not recorded.

## Case 1

An 80-year-old man, who had been living in the UK for 49 years, was diagnosed with BT leprosy having developed red plaques with reduced sensation and thickened peripheral nerves one week after a second dose of BNT162b2 vaccine. The diagnosis was confirmed on skin biopsy with peri-neural and peri-adnexal granulomatous inflammation with infiltration and destruction of dermal nerves. The Wade-Fite stain was negative. Slit-skin smear was negative. The skin lesions and nerve thickening improved noticeably within eight weeks of anti-microbial therapy. Interestingly he had a third dose of BNT162b2 vaccine six months after the second dose having started anti-bacterial therapy and experienced no deterioration of his leprosy. The skin lesions and nerve thickening had resolved by the time he completed the six-month course of anti-microbial therapy. There had been no recurrence of the plaques or nerve signs after 12 months.

## Case 2

A 27-year-old man who had taken eight months of multibacillary MDT more than three years earlier developed red plaques and tender, thickened nerves consistent with a leprosy T1R eight weeks after a single dose of BNT162b2 vaccine and prior to arrival in the UK. He had received a second of two CoronaVac vaccine three months before the BNT162b2 vaccine. The diagnosis was supported by a skin biopsy which showed non-necrotising destructive granulomatous neuritis, oedema and epidermal HLA-DR up-regulation consistent with a T1R. The anti-BCG immunostain was positive although Wade-Fite stain was negative. The skin lesions and tender nerves improved with a reducing course of prednisolone and had not recurred after 12 months.

## Discussion

Our retrospective cohort study showed a very high (98%) uptake of SARS-CoV-2 vaccine in individuals attending the Leprosy Clinic at HTD in the UK.

SARS-CoV-2 vaccination was associated with the development of leprosy in one individual and a T1R in another. Both men developed their leprosy adverse reaction following vaccination with BNT162b2 although in each case it was not the first SARS-CoV-2 vaccination either had received. This may be due to increased TNF-α and interleukin-6 (IL-6) after BNT162b2 second doses [30].

We identified 14 individuals, men, and women, with leprosy adverse events associated with SARS-CoV-2 vaccines in six published reports from both leprosy endemic and non-endemic settings. These reports are summarised in Table 3.

**Table 3. Cases of leprosy adverse effects associated with SARS-CoV-2 vaccine.**

| Country of origin of report | Sex | Age (years) | Vaccine | Number of vaccines prior to onset of adverse event | Leprosy adverse event | Timing of adverse event after most recent vaccine (Days) | MDT status at time of vaccination | Treatment |
|---|---|---|---|---|---|---|---|---|
| **India** [31] | Female | 35 | ChAdOx1 nCoV-19 | 1 | T1R complicating BT leprosy | 14 | None | WHO MB-MDT Prednisolone |
| **India** [32] | Male | 35 | ChAdOx1 nCoV-19 | 1 | T1R | 11 | Current | Prednisolone |
| **Indonesia** [33] | Male | 33 | Not stated | 1 | T1R | 14 | Current | Prednisone |
| **Brazil** [34] | Male | 44 | ChAdOx1 nCoV-19 | 1 | ENL | 3 | Completed | Thalidomide |
| **Brazil** [34] | Male | 43 | ChAdOx1 nCoV-19 | 1 | ENL complicating relapse/reinfection | 2 | Completed | Thalidomide WHO MB-MDT |
| **India** [35]* | Male | 60 | ChAdOx1 nCoV-19 | 1 | Recurrence of ENL | 5 | Completed | Not stated |
| **India** [31] | Female | "40s" | ChAdOx1 nCoV-19 | 1 | ENL | 7 | Completed | Prednisolone |
| **India** [31] | Male | 45 | ChAdOx1 nCoV-19 | 1 | ENL | 8 | Completed | Prednisolone |
| **Israel** [36] | Male | 32 | BNT162b2 | 1 | ENL complicating undiagnosed LL | 14 | None | WHO MB-MDT |
| **Singapore** [37] | Male | 24 | BNT162b2 | 1 | Leprosy reaction type not specified complicating undiagnosed lepromatous disease | 10 | None | Clarithromycin, minocycline and ofloxacin Prednisolone |
| **India** [32] | Male | 28 | ChAdOx1 nCoV-19 | 2 | T1R | 5 | Completed | Prednisolone |
| **India** [32] | Male | 25 | ChAdOx1 nCoV-19 | 2 | T1R | 8 | Completed | Prednisolone |
| **India** [32] | Male | 45 | BBV152 COVAXIN | 2 | ENL | 7 | Current | Thalidomide |
| **Taiwan** [38] | Male | 36 | mRNA-1273 | 3 | T1R | 7 | Current | Methylprednisolone Methotrexate |
| **UK**** | Male | 80 | BNT162b2 | 2 | BT leprosy | 7 | None | mROM§ |
| **UK**** | Male | 27 | BNT162b2 | 3 | T1R | 56 | Completed | Prednisolone |

*individual had a second episode 5 days after a second dose of SARS-CoV-2 vaccine

** current study, § monthly rifampicin, ofloxacin and minocycline

The onset of reported leprosy adverse events were 5–14 days following a SARS-CoV-2 vaccine. The onset of T1R in our second case was 56 days after his third SARS-CoV-2 vaccine. This is longer than other reported cases but may reflect a lack of recognition of the potential association in individuals who have a longer duration between vaccination and onset of the leprosy adverse event. Ten of the 14 individuals experienced a leprosy adverse event after the first vaccination, three after the second and one after a third vaccine. All four new presentations of leprosy were associated with reactions unlike our first case. Only one of the four cases of leprosy associated with a SARS-CoV-2 vaccine, that of the 35-year-old woman with BT leprosy and T1R reported by Bhandari and colleagues [31] could be a new manifestation of leprosy associated with vaccination. The other three were highly smear positive indicating a longstanding *M. leprae* infection predating vaccination [39].

Indeterminate, tuberculoid and borderline tuberculoid leprosy without evidence of reaction occurred in 17 of the 21 (81%) previously healthy individuals who were diagnosed with leprosy 8–12 weeks following BCG vaccination compared to four with T1R [40].

Six cases of T1R and seven cases of ENL were reported. Aponso and colleagues did not specify the type of reaction in their highly smear positive untreated patient [37]. The reports of associations between immune mediated inflammatory complications of leprosy such as T1R, ENL and neuritis and SARS-CoV-2 vaccination is compatible with the T cell, cytokine and other immune responses elicited by the vaccines. In our cohort, 69% of patients had completed MDT more than three years prior to vaccination and were at low risk of new onset leprosy reactions. The Uniform MDT study from Brazil showed that leprosy reactions are rare events three years after starting MDT [41]. This strengthens the likelihood of an association between vaccination and the development of T1R in our second case, who had taken MDT more than three years prior to receiving his third SARS-CoV-2 vaccine.

Interestingly two individuals who developed ENL following a first SARS-CoV-2 vaccine were able to have a second dose sixteen weeks later without ill effect [31].

A systematic review by Avallone et al. [42] showed the wide range of severe inflammatory dermatological conditions following SARS-CoV-2 vaccination but concluded they were "...infrequent, and not life-threatening". A similar review of peripheral nervous system adverse events following mRNA SARS-CoV-2 vaccines showed vaccinated individuals had a higher risk of Bell's palsy than unvaccinated which the authors described as "marginal" [43].

The Indian Association of Dermatologists, Venereologists and Leprologists Special Interest Group on leprosy recommend that SARS-CoV-2 vaccination should not be delayed in individuals with leprosy [44]. Large scale campaigns were undertaken to ensure that individuals affected by leprosy had access to vaccination [45].

It is important for clinicians to be aware of the potential leprosy adverse events associated with SARS-CoV-2 vaccination but given the small numbers of reports leprosy should not be considered a contraindication to vaccination. Work to understand high reported rates of acceptability of vaccination among people affected by leprosy may provide insights into work to address vaccine hesitancy. The longstanding association between vaccination and the development of leprosy *per se* or leprosy reactions needs to be borne in mind when developing and testing therapeutic vaccines for leprosy [46].

## Supporting information

**S1 Data. Database.**
(XLS)

## Acknowledgments

We are grateful to the patients who attend the Leprosy Clinic at the Hospital for Tropical Diseases, London for their contribution to the study.

## Author Contributions

**Conceptualization:** Stephen L. Walker.

**Data curation:** Barbara de Barros.

**Project administration:** Barbara de Barros, Stephen L. Walker.

**Resources:** Barbara de Barros, Rachel Pierce, Cathryn Sprenger, Eugene Liat Hui Ong, Stephen L. Walker.

**Supervision:** Stephen L. Walker.

**Writing – original draft:** Barbara de Barros.

**Writing – review & editing:** Barbara de Barros, Rachel Pierce, Cathryn Sprenger, Eugene Liat Hui Ong, Stephen L. Walker.

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
