## [Decision Letter · Decision Letter 0]

26 May 2023

Dear BaBaDr de Barros,

Thank you very much for submitting your manuscript "COVID-19 vaccination and leprosy – a UK hospital-based retrospective cohort study" for consideration at PLOS Neglected Tropical Diseases. As with all papers reviewed by the journal, your manuscript was reviewed by members of the editorial board and by several independent reviewers. The reviewers appreciated the attention to an important topic. Based on the reviews, we are likely to accept this manuscript for publication, providing that you modify the manuscript according to the review recommendations. 

Sincerely,

Linda B Adams

Academic Editor

Elsio Wunder Jr

Section Editor

Reviewer's Responses to Questions

**Key Review Criteria Required for Acceptance?**

**Methods**

-Are the objectives of the study clearly articulated with a clear testable hypothesis stated?

-Is the study design appropriate to address the stated objectives?

-Is the population clearly described and appropriate for the hypothesis being tested?

-Is the sample size sufficient to ensure adequate power to address the hypothesis being tested?

-Were correct statistical analysis used to support conclusions?

-Are there concerns about ethical or regulatory requirements being met?

Reviewer #1: - The objectives of the study are not explicitly stated in the text. However, the study's aim appears to be to investigate whether individuals with leprosy who received the COVID-19 vaccine experienced any adverse events, including the development of leprosy or leprosy reactions within twelve weeks of vaccination. The study's findings suggest that vaccine-mediated T cell responses may be associated with the development of these adverse events in individuals with leprosy or latent M. leprae infection.

-yes, it is appropriate however the study's sample size is very small, the findings suggest a possible association between vaccine-mediated T cell responses and the development of leprosy or leprosy reactions in individuals with leprosy or latent M. leprae infection.

-yes, it is clearly described. 

-sample size is small and not sufficient to address the hypthesis being tested. 

-Did not find any statistical analysis plan - simple analysis of the data was done

-not much of a concern with ethics as it is a retrospective study and data was taken from EMR. Data collection was approved as part of a quality improvement project registered with the Infection Division of University College London Hospitals NHS Foundation Trust in accordance with institutional policy.

Reviewer #2: Methods are clearly articulated and design is appropriate to test the hypothesis presented.

The population is clearly described and is appropriate. Sample size is not relevant as this study is analysing a retrospective cohort.

No statistical analysis is used beyond expressing data in percentages.

There was apparently no attempt to obtain individual informed consent, but if all data was extracted from clinical records and anonymised by the team responsible for clinical care of patients, in conducting this Quality Improvement exercise, then it would be reasonable not to seek individual’s consent. See lines 151-154:data extraction approved in line with institutional policy. No identifiable personal data is included and no photographs.

Reviewer #3: The submission intends to describe leprosy reactions to occur as a potential consequence of vaccination against SARS-CoV-2. Leprosy clinic attendees were evaluated, and the study is not powered (nor is it presented as such) to support any statistical evaluation.

**Results**

-Does the analysis presented match the analysis plan?

-Are the results clearly and completely presented?

-Are the figures (Tables, Images) of sufficient quality for clarity?

Reviewer #1: -no plan was presented 

-yes, results presented clearly

-yes

Reviewer #2: Result are presented clearly according to the analysis plan. There are no figures, the 2 tables are sufficiently clear.

Their findings are discussed in relation to other published reports of similar observations from endemic countries

Reviewer #3: The authors reviewed clinical records of 52 clinic attendees in 2021, reporting that 2 had leprosy or reactional episodes after receipt of the BNT162b2 vaccine (1 one week after second dose, 1 eight weeks after a single dose)

**Conclusions**

-Are the conclusions supported by the data presented?

-Are the limitations of analysis clearly described?

-Do the authors discuss how these data can be helpful to advance our understanding of the topic under study?

-Is public health relevance addressed?

Reviewer #1: -yes, conclusions are supported by the data presented 

-The authors have clearly described the limitations of their analysis, such as the small sample size and lack of control group, which may limit the generalizability of their findings.

-Yes They have also discussed how their data can be helpful in advancing our understanding of the topic under study, specifically the potential association between COVID vaccination and leprosy adverse events.

-The authors have addressed the public health relevance of their findings by emphasizing the importance of clinicians to be aware of the potential leprosy adverse events associated with COVID vaccination. They have also highlighted the need for large-scale campaigns to ensure that individuals affected by leprosy have access to vaccination. 

Vaccination should not be delayed in individuals with leprosy.

Reviewer #2: The cautious conclusions are justified by the findings described. The authors explain potential relevance of findings, in relation to individuals’ and public health.

Reviewer #3: Conclusions are simple statements of the observations, which is appropriate given the limited data set. While attention is drawn to the 2 episodes that are clinically suggested to the related to receipt of BNT162b, and appropriate reference is made to the impact of BCG immunization on leprosy reactions, it is unclear if the observation is simply related to vaccination in general or to the COVID-19 vaccine. This reservation should be made more clear. Consideration of records and impacts of other vaccines (i.e., shingles) that might be commonly administered to this population should also be made. Further, review of historical records (pre-COVID) would provide an indication as to if 2/52 attendees an increase over typical presentation rates.

**Editorial and Data Presentation Modifications?**

Reviewer #1: Accept

Reviewer #2: Presentation does not require changing.

Reviewer #3: (No Response)

**Summary and General Comments**

Reviewer #1: This study highlights a potential association between Covid vaccination and the onset of leprosy reactions and provides detailed clinical information on the cases and their outcomes. There is a comparison of findings with previous literature on vaccination and leprosy, which helps put the results in context. The study raises awareness among clinicians to consider the potential association between vaccination and leprosy reactions.

There was no control group, and the results may be subject to bias and the sample size was small, with only 14 cases identified, which limits the generalizability of the findings. 

The novelty of the study is that it reports a possible association between vaccination and leprosy reactions, specifically T1R and ENL the first study to do so. 

The study also highlights the importance of monitoring potential adverse events following vaccination and the need for further research to understand the mechanisms behind these associations.

The evidence may be of interest to the general public as it explores a potential association between leprosy adverse events and Covid vaccination. It provides important information for healthcare professionals and individuals affected by leprosy who are considering vaccination against COVID-19. However, the study has limitations, and the reported cases are rare, so it should be interpreted with caution.

Reviewer #2: This is an interesting report of a retrospective study of covid-19 vaccination status in leprosy affected people attending a tertiary care facility in a non-endemic country. Possibly different findings might result from a similar study in a highly endemic country with different vaccination practices

Reviewer #3: The report is a straightforward presentation attempting to link COVID-19 vaccination with leprosy reactions. Wider consideration of historical trends and receipt of other vaccines would significantly elevate the study and test the hypothesis more fully.

PLOS authors have the option to publish the peer review history of their article (what does this mean?). If published, this will include your full peer review and any attached files.

Reviewer #1: Yes: J.Darlong

Reviewer #2: No

Reviewer #3: Yes: Malcolm Duthie

Figure Files:

Data Requirements:

Reproducibility:

References

---

## [Editor Report · Decision Letter 1]

1 Jul 2023

Dear BaBaDr de Barros,

We are pleased to inform you that your manuscript 'COVID-19 vaccination and leprosy – a UK hospital-based retrospective cohort study' has been provisionally accepted for publication in PLOS Neglected Tropical Diseases.

Best regards,

Linda B Adams

Academic Editor

Elsio Wunder Jr

Section Editor

---

## [Editor Report · Acceptance letter]

1 Aug 2023

Dear BaBaDr de Barros,

We are delighted to inform you that your manuscript, "COVID-19 vaccination and leprosy – a UK hospital-based retrospective cohort study," has been formally accepted for publication in PLOS Neglected Tropical Diseases.

Best regards,

Shaden Kamhawi

co-Editor-in-Chief

Paul Brindley

co-Editor-in-Chief
